# Higher Impulse Electromyostimulation Contributes to Psychological Satisfaction and Physical Development in Healthy Men

**DOI:** 10.3390/medicina57030191

**Published:** 2021-02-25

**Authors:** Kangho Kim, Denny Eun, Yong-Seok Jee

**Affiliations:** Research Institute of Sports and Industry Science, Hanseo University, Seosan 31962, Korea; kanghokim.aiden@gmail.com (K.K.); eun23415@gmail.com (D.E.)

**Keywords:** electromyostimulation, body image, body shape, self-esteem, body composition

## Abstract

*Background and Objectives*: This study investigated the various impulse effects of whole-body electromyostimulation (WB-EMS) on psychophysiological responses and adaptations. *Materials and Methods*: The participants included fifty-four men between 20 and 27 years of age who practiced isometric exercises for 20 min, three days a week, for 12 weeks while wearing WB-EMS suits, which enabled the simultaneous activation of eight muscle groups with three types of impulse intensities. Participants were allocated to one of four groups: control group (CON), low-impulse-intensity group (LIG), mid-impulse-intensity group (MIG), and high-impulse-intensity group (HIG). Psychophysiological conditions were measured at week 0, week 4, week 8, and week 12. *Results*: Compared with the CON, (1) three psychological conditions in LIG, MIG, and HIG showed positive tendencies every four weeks, and the analysis of covariance (ANCOVA) test revealed that body image (*p* = 0.004), body shape (*p* = 0.007), and self-esteem (*p* = 0.001) were significantly different among the groups. (2) Body weight, fat mass, body mass index, and percent fat in the CON showed decreasing tendencies, whereas those in LIG, MIG, and HIG showed a noticeable decrease, which revealed that there were significant differences among the groups. Specifically, a higher impulse intensity resulted in a greater increase in muscle mass. (3) Although there was no interaction effect in the abdominal visceral fat area, there were significant interactions in the abdominal subcutaneous fat (ASF) and total fat (ATF) areas. Both the ASF and ATF in the CON showed decreasing tendencies, whereas those in other groups showed a noticeable decrease. The ANCOVA revealed that the ASF (*p* = 0.002) and ATF (*p* = 0.001) were significantly different among the groups. In particular, the higher the impulse intensity, the greater the decrease in abdominal fat. *Conclusions*: This study confirmed that high-impulse-intensity EMS can improve psychophysiological conditions. In other words, healthy young adults felt that the extent to which their body image, body shape, and self-esteem improved depended on how intense their EMS impulse intensities were. The results also showed that higher levels of impulse intensity led to improved physical conditions.

## 1. Introduction

The definitions of health and beauty have varied over time, but the general standard for men has always been an image of being slim and muscular [1]. As with many issues, body image has become a concern in our society. Body image is the perception of one’s body size and appearance and the emotional responses to this perception [2]. Cash [3] reported that body image is a multidimensional construct and refers to a person’s perceptions and attitudes, including feelings, thoughts, and behaviors regarding their own body and appearance. Cash [4] also reported that the cognitive-behavioral model of the body image, which includes personality, physical or interpersonal attributes, and cultural socialization, plays a role in how invested individuals are in their body image and how they evaluate it. One facet of attitudinal body image is referred to as body satisfaction or dissatisfaction [5]. Inaccurate perceptions of body size and negative emotional reactions can result in varying degrees of body image dissatisfaction. Negative views towards obesity have been internalized. Many people have adopted the belief that obese individuals are unattractive, psychologically impaired, or medically sick [6]. Obesity caused from a sedentary lifestyle is associated with inappropriate food intake and energy imbalances [7]. Among the kinds of obesity, abdominal obesity is mainly seen in males and is a dangerous factor that causes heart and metabolic diseases, as well as blood vessel problems. Moreover, the larger number of adipocytes in the abdominal region increases metabolic complications [8,9]. This association seems to be due to a higher lipolytic rate in the visceral and deep subcutaneous adipose tissue, promoting an increase of free fatty acids in the blood circulation [10] and an increase in the hepatic synthesis of triglycerides, which translates into dyslipidemia [11]. Additionally, adipose tissue plays an important role in the development of systemic inflammation by secreting several cytokines and chemokines [12,13].

Physical activity, controlled diet, anti-obesity medication, and liposuction represent significant modalities in the treatment of obesity, resulting in increased energy expenditure, decreased energy uptake, reduced fat tissue, and an increased lean body mass. Additionally, regular physical activity and exercise have long been known to increase the metabolism and reduce fat mass, contributing to a more positive body image or shape [14,15], as well as self-esteem [16]. However, attempts to resolve obesity through traditional exercise or diets are somewhat inadequate for people who want fast results or those with metabolic syndromes. In addition, if abdominal obesity cannot be resolved within a short period of time, various complications can result, such as those described above. Therefore, other nonconventional methods have been devised.

Whole-body electromyostimulation (WB-EMS) is a somewhat newly adopted device that provides exercise-like effects in which artificial contractions are induced by electric currents from an external source [17,18], unlike natural contractions induced by the motor nerve of the central nervous system. Electrical muscle stimulation (EMS) delivers a stimulus to local muscles in a static state at sufficient intensities to evoke muscle contractions [19]. WB-EMS is time-efficient and less debilitating than localized EMS, thus producing a higher acceptance among nonathletes or athletes [20]. Maffiuletti [21] suggested that electrical stimulation increases the maximal strength and improves physical fitness. The authors von Stengel and Kemmler [22] analyzed the changes in the maximum isokinetic leg/hip extensor strength and leg/hip flexor strength after WB-EMS interventions in men from different periods of life. They found that, although there was an inconsistent tendency in terms of WB-EMS-induced lower extremity strength, WB-EMS significantly increased the maximal hip/leg strength throughout the adult male lifespan. Furthermore, Kemmler et al. [23] demonstrated that WB-EMS had positive effects on muscle mass and fat mass, as well as improved functional capacity, even in older, sedentary people. They also reported that WB-EMS is gentle on the joints and reduces the risk of injury due to excessive loads resulting from weight training. Recently, Kim and Jee [24] reported that obese elderly women who exercised with music while wearing WB-EMS suits resulted in improved body composition and cholesterol levels after eight weeks. In addition, they also found that there were decreased tendencies in some cytokines such as tumor necrosis factor-a, C-reactive protein, resistin, and carcinoembryonic antigen in the WB-EMS intervention group.

However, even though there is some evidence that WB-EMS favorably improves the body composition, biomarkers, and muscle mass or strength, few studies clearly address those benefits. Particularly, it has not been confirmed that a dose-response relationship exists between different impulse intensities and how WB-EMS affects the psychological factors such as body image or shape and self-esteem. Therefore, this study investigated the various impulse effects of WB-EMS on psychological conditions (body image, body shape, and self-esteem) and physical conditions (body composition and abdominal fatness) in healthy young men in accordance with dose responses using electrical stimulation composed of different impulse intensities.

## 2. Materials and Methods

### 2.1. Participants

The participants were aged between 20 and 27 years old. All volunteers wanted to improve their body shape and were checked by a bioelectrical impedance analysis (BIA) device. This study recruited healthy male college students who lived in a dormitory and did not exercise regularly for a duration of six months. They were excluded if they received any treatment for weight loss, taken any medication known to affect the body composition, or underwent any major surgery during one year prior to the start of the study. The following were also reasons for exclusion: having a history of coronary arterial disease, severe cerebral trauma, cerebrovascular disease, pulmonary disease, uncontrolled hypertension, cancer, and psychiatric diseases, such as eating disorders. After completing a survey and taking baseline measurements, fifty-four participants were randomly assigned to one of four groups using random number tables and assigned identification numbers upon recruitment: the control group (CON, *n* = 13), low-impulse-intensity group (LIG, *n* = 13), mid-impulse-intensity group (MIG, *n* = 14), and high-impulse-intensity group (HIG, *n* = 14). Although sixty participants were initially gathered, three participants were excluded, because two of them took part in an exercise program for over six months and another refused to participate. In the follow-up phase and the analysis phase, two participants from the MIG and one participant from the HIG dropped out due to personal reasons. Finally, fifty-four participants took part in this study, as shown in Table 1.

### 2.2. Experimental Design

This single-blind, randomized, controlled trial was conducted in a research center at Hanseo University, Seosan, Korea. This study followed the principles of the Declaration of Helsinki and received approval from the institutional ethics committee (26 September 2017 to 25 September 2018; 2-1040781-AB-N-01-2017083HR). This study was registered with Clinical Research Information Service (CRIS), reference KCT0005931. Prior to the study, the principal investigator explained all the procedures to the participants in detail. All participants read and signed an informed consent form. They arrived at the research center to complete a self-reported questionnaire about their health status and to learn how to record their calorie intake and calorie output in a diary.

The intervention program of this study lasted for 12 weeks, similar to the duration of previous studies [25,26,27]. The assessments were performed at week 0 (baseline) and then every 4 weeks for 12 weeks. Although all participants were assigned to four groups, no participant or other staff members were aware of the group assignments for the duration of the trial. All participants wore WB-EMS suits that fit their individual size. The LIG, MIG, and HIG underwent 20-min WB-EMS sessions combined with isometric exercises in accordance with their intensities of electrical stimulation 3 times a week for 12 weeks. In other words, they received one of three types of electrical stimuli at different intensities according to their maximum tolerance (1 MT). The CON also wore the WB-EMS suit as much as the other groups, but they did not receive any electrical stimuli while performing isometric exercises. The amount and intensity of the isometric exercise was the same for all the groups.

### 2.3. Measurement Methods

#### 2.3.1. Measurement of Calorie Intake and Calorie Output

At the pre-experiment session, the participants were provided a diary to record what they consumed for breakfast, lunch, and dinner throughout the 12-week experimental period. However, for the record of calories consumed at week 0, the foods consumed on the day before the experiment were recorded in the diary. An expert input the food type and volume into CAN-Pro 5.0 (Korean Nutrition Society, Seoul, Korea) every day, calculated the caloric intake, and then performed an evaluation at the end of each month. The recorded calorie intake data were averaged and analyzed at the baseline, week 4, week 8, and week 12.

The daily amount of physical activity that was performed outside the experiment was recorded and calculated using the international physical activity questionnaire (IPAQ)—shortened form version [28,29]. In order to increase the accuracy of the responses, an expert provided a diary to record the contents of the questionnaire on a daily basis. The participants answered the questionnaires based on the recordings of physical activities for the past 7 days throughout the 12-week experimental period. For the record of calorie output for week 0, physical activity performed on the day before the experiment was recorded in the diary. The daily calorie output was calculated by metabolic equivalent (MET)/minutes (kcal/kg/min) at the end of every month (Table 2).

By using Table 2, the total score was obtained through the summation of the duration (in minutes) and frequency (days) of walking, moderate-intensity activity, and vigorous-intensity activity. Then, using the data, the average amount of physical activity per week was calculated based on the IPAQ score conversion method. These data were averaged on a monthly basis and analyzed every 4 weeks.

#### 2.3.2. Measurement of Psychological Conditions

The psychological conditions for healthy male participants were evaluated and analyzed in three categories, and the reliabilities of three questionnaires from week 0 to week 12 were measured by calculating Cronbach’s α, representing internal consistencies, as shown in Table 3. First, the Body Image Acceptance and Action Questionnaire (BI-AAQ) was used to measure the participants’ flexibility and acceptance regarding their own body image [30]. This questionnaire was developed to measure body flexibility that assesses the acceptance of negative or unwanted thoughts, perceptions, body sensations, and emotions related to their body. The BI-AAQ was comprised of 12 items rated on a 7-point scale ranging from 1 = never true to 7 = always true. The participants’ scores on the items were reverse-scored and averaged; lower scores reflected higher levels of body image flexibility. 

Second, the Body Shape Questionnaire (BSQ) developed by Cooper et al. [31], an 8-item self-reported measure, was used to assess negative concerns regarding body shape and size. This questionnaire consisted of four factors: fear of obesity, fear of exposure, experience of vomiting, and body dissatisfaction. Questions such as “Have you pinched areas of your body to see how much fat there is?” and “Have you thought that your thighs, hips, or bottom are too large for the rest of you?” were measured on a 6-point scale (1 = never to 6 = always). A lower total score was indicative of greater body satisfaction.

Third, the self-esteem scale (SES) of Rosenberg was used to measure the self-esteem of college students [32]. This questionnaire was designed to conceptualize self-esteem as a single dimension and to allow the participant to comprehensively evaluate oneself. The items of the survey instrument measured the degrees of the self-esteem and self-approval patterns, which consisted of 10 items, with 5 items regarding positive self-esteem (questions 1, 2, 3, 4, and 5) and 5 items regarding negative self-esteem (questions 6, 7, 8, 9, and 10). Negative self-esteem was scored as a reverse item. A higher total score represented a higher level of self-esteem.

#### 2.3.3. Measurement of Physiological Conditions

The physiological conditions for the participants of this study were evaluated and analyzed by the BIA and computer tomography (CT) methods as follows. First, regarding the BIA, height was measured using a BMS 330 Anthropometer (Biospace Co., Ltd., Seoul, Korea), and body weight, muscle mass, fat mass, and percent fat of the participants were assessed using an InBody 230 Body Composition Analyzer (Biospace Co., Ltd., Seoul, Korea). This analyzer is a segmental impedance device, in which the electrodes are made of stainless-steel interfaces. The participants stood upright by placing their feet on the foot electrodes and gripping the hand electrodes. Eight tactile electrodes were attached to the surfaces of both hands and feet: palms, fingers, front soles, and rear soles. An analysis of body composition was measured before dinner and after voiding [33].

Second, in the aspect of CT screening, all participants of the study visited the Seoul Songdo Hospital in Korea. The participants lay down horizontally with their face and torso facing up, with both arms raised overhead. A radiologist performed a CT scan (Toshiba Scanner Aquilion Prime Model TSX-303A, Toshiba Medical Systems Corporation, Tokyo, Japan) of the abdomen four times (week 0, week 4, week 8, and week 12). All measurements were taken by a radiologist throughout the study for minimizing the measurement errors. This scan for abdominal fatness was performed at the level of the umbilicus or between the 4th and 5th lumbar vertebra. Abdominal visceral fat (AVF) and abdominal total fat (ATF) areas were estimated by delineating the regions and calculating an attenuation range of −190 to −30 Hounsfield units. The abdominal subcutaneous fat (ASF) area was calculated by subtracting the AVF area from the ATF value. The unit of all the area values was cm^2^. This study tried to secure the safety of the participants by measuring the abdominal circumference in the shortest time while minimizing the radiation dose from the CT scans [34,35]. In addition, the exposure dose of the participants was measured. An average value of 1.69 mSv was measured from the abdominal fat CT scans for a total average of about 6.91 mSv from all four measurements. These results were somewhat higher than natural radiation exposure (about 3 mSv) and radiation dose (about 3–7 mSv) that airplane crews are typically exposed to during one year. However, it was lower than the annual radiation dose of radiation workers (20 mSv per year) or the average radiation dose (about 1000–2000 mSv) received during radiation therapy for cancer treatment. In other words, it was considered and confirmed that this dose was not dangerous or harmful to the participants’ health [36].

#### 2.3.4. Measurement of Creatine Kinase

This study also took the safety of the participants into consideration by measuring the creatine kinase (CK). In other words, the CK was included, because it was considered to be an indicator of muscle damage during or after exercise with electrical muscle stimulation. Blood samples were taken after fasting for 10 h or longer before assessment and were collected using BD vacutainer tubes (Becton Dickinson, Franklin Lakes, NJ, USA) at 8 a.m. the following day. After the participants were stabilized for 10–15 min, 5 mL of blood was collected from the antecubital vein of the participants with a disposable syringe by a medical laboratory technologist. A total of 2 mL of the 5 mL of venous blood was added to an anticoagulant tube (EDTA bottle), shaken, and centrifuged at 3000 rpm for 5 min. The remaining 3 mL was left at room temperature for 1 h and centrifuged at 1000 rpm for 15 min. Isolated serum was kept frozen until the test. The samples were taken to the laboratory for analysis [37]. The CK was analyzed using a Beckmann Coulter Inc. device (Brea, CA, USA) at week 0, week 4, week 8, and week 12.

### 2.4. WB-EMS Protocol

Participants were given variously sized WB-EMS suits made by Miracle^®^ (Seoul, Korea) according to their size. The suit was composed of a silicone conductive pad and controlled via Bluetooth. This suit enabled the simultaneous activation of eight pairs of muscle groups (upper legs, upper arms, buttocks, abdomen, chest, lower back, upper back, and latissimus dorsi) with selectable intensities for each region [24]. In order to generate the effects of a diverse range of motion, eight types of isometric movements were performed during the impulse phase, as per the instructor’s direction, as shown in Figure 1. Based on the available literature [18,38,39,40,41], the stimulation frequency was selected at 85 Hz, the impulse width at 350 microsec, the impulse rise as a rectangular application, and the impulse intensity as a relative voltage on the maximal peak voltage (160 V). The impulse duration was 6 s, with a 4-s break between impulses. Each group trained by a qualified instructor conducted 20-min WB-EMS sessions 3 times a week (Mondays, Wednesdays, and Fridays) on two nonconsecutive days to allow a rest between each session.

This study used 1 MT as the maximum peak voltage, similar to calculating the maximal voluntary contraction as one maximal repetition [42]. Each 1 MT of the upper and lower body was measured and stored via Bluetooth, and the intensity was adjusted for each individual during the isometric exercise. For preventing the patients from being surprised or uncomfortable with the electrical stimulus, the 1-MT level was gradually increased after starting with a low stimulation current [43,44,45]. All patients were asked to express the difficulty level of the exercise during the isometric exercises with wearing an EMS suit. An instructor asked for the ratings of perceived exertion (RPE) every 5 min, and an assistant recorded them. The electric stimulation was stopped at the request of the participant when reaching an unbearable level on the RPE scale [46]; at which point, the intensity was set as 1 MT. In other words, the % MT of this study was obtained through the RPE scale, a numerical scale ranging from 6 to 20, where 6 means “no exertion at all” and 20 means maximal exertion. The intensity of the exercise was estimated by applying the RPE during the exercise to the CON, as well as the three experimental groups. The intensity of the electrical workout was different from 1 MT. The LIG was assigned to 60% of 1 MT, MIG to 70% of 1 MT, and HIG to 80% of 1 MT from baseline to the end of the experiment. Although the CON performed isometric exercises while wearing EMS suits, they did not receive any electrical stimuli.

### 2.5. Statistical Analyses

Microsoft Excel (Microsoft, Redmond, WA, USA) was used to analyze the data, expressed as mean ± standard deviation (SD). The sample size was determined using G*Power v. 3.1.9.7 [47,48], considering a priori effect size of f2 (V) = 0.25 (medium size effect), α error probability = 0.05, power (1-β error probability) = 0.95, number of groups = 4, and number of measurements = 4. There were 13 to 14 subjects that were assigned to each of the 4 groups, with a total of 52 subjects based on the numbers assigned to this program. SPSS (version 22.0; IBM Corp., Armonk, NY, USA) was used to perform all statistical analyses, and the Shapiro–Wilk test was used to check the data distribution. Differences between the groups were observed using the Kruskal-Wallis rank test prior to comparing the groups. An analysis of variance (ANOVA) test was used for evaluating significant variances between the groups at the baseline, and 4 × 4 (group, time, and group by time interaction) was used to assess the effects of intervention. An analysis of covariance (ANCOVA) test was used to determine the differences between groups if there was an interaction between group and time (pre-values and post-values). Moreover, the Bonferroni post hoc test was implemented if there were significant differences among the four groups. An intention-to-treat analysis was conducted to compare the CON, LIG, MIG, and HIG. The groups served as the between-group factor, and the week 0 vs. week 4 vs. week 8 vs. week 12 were the within-group factors. For all analyses, the significance level was set at *p* ≤ 0.05.

## 3. Results

### 3.1. Demographic and Controlled Variables

As shown in Table 1, there were no significant differences among the four groups. The demographic variables of this study indicated a homogeneity of subjects. There were also no significant differences in the controlled variables, as shown in Table 4.

### 3.2. Effects of WB-EMS Exercise on Psychological Conditions

There were significant interaction effects for all psychological questions (Table 5). Three psychological scales in the CON showed negative changing tendencies, whereas those in the other groups showed positive changing tendencies. The ANCOVA revealed that the BI-AAQ (*F* = 5.017, *p* = 0.004), BSQ (*F* = 4.680, *p* = 0.007), and self-esteem questionnaire (SEQ) (*F* = 8.468, *p* = 0.001) were significantly different among the four groups (not shown in Table 5). In particular, the HIG showed the most improved value in week 12, which was confirmed by the Bonferroni post hoc test.

### 3.3. Effects of WB-EMS Exercise on Physiological Conditions

As shown in Table 6, weight, fat mass, BMI, and percent fat in the CON showed decreasing tendencies, whereas those in the LIG, MIG, and HIG showed a noticeable decrease. There were significant interactions in all the variables in the repeated ANOVA test. The ANCOVA test revealed that, although weight (*F* = 6.354, *p* = 0.001), fat mass (*F* = 7.368, *p* = 0.001), and BMI (*F* = 6.427, *p* = 0.001) in the three experimental groups were significantly lower than those in the CON, the percent fat (*F* = 2.268, *p* = 0.092) did not show a significant difference. Muscle mass showed a different tendency, and the ANCOVA revealed that muscle mass (*F* = 5.758, *p* = 0.002) in the three experimental groups was significantly higher than those in the CON. In particular, the higher the impulse intensity was applied, the greater the increase of muscle mass.

### 3.4. Effects of WB-EMS Exercise on Abdominal Fatness

Although there was no interaction effect in the abdominal visceral fat area, there were significant interactions in the abdominal subcutaneous fat and total fat areas (Table 7). Both subcutaneous fat and total fat in the CON showed decreasing tendencies, whereas those in the experimental groups showed a noticeable decrease. The ANCOVA revealed that subcutaneous fat (*F* = 5.517, *p* = 0.002) and total fat (*F* = 10.933, *p* = 0.001) were significantly different among the groups. In particular, the HIG showed the lowest value in week 12, which was confirmed by the Bonferroni post hoc test.

## 4. Discussion

This study found that three psychological scales in the CON showed insignificant or negative changing tendencies, whereas those in the LIG, MIG, and HIG showed positive changing tendencies. Furthermore, the BI-AAQ for body image, BSQ for body shape, and SEQ for self-esteem were significantly different among the groups, which showed that higher impulse intensities resulted in greater positive changes. In other words, the HIG, which received the highest impulse intensity, showed the most improved values from week 0 to week 12. As for body composition and psychological variables, the WB-EMS groups, which were given stronger stimulations, improved their body weight, fat mass, and muscle mass, especially in the ASF and ATF. The physiological variables also showed a positive relationship between higher impulse intensities and a greater degree of improvement.

Everyone knows that exercise leads to physical development. Particularly, exercise can provide greater benefits when the intensity is higher than that of daily physical activity. However, if the intensity of exercise is too high or excessive, it can cause severe damage to stressed joints, as well as muscle ruptures. The WB-EMS suit, which has been used since several years ago to compensate for this, protects the muscle joints of the human body by reducing the burden caused by the weight from isotonic exercise but can maximize the effect of exercise by increasing the intensity of exercise. The effects of the high WB-EMS impulse intensity used in this study were similar to the results of another research study that showed increased lipid oxidation leading to positive effects on the metabolic indicators and body composition in obese men [49]. This study also showed that electrical current thresholds were higher in obese than in nonobese subjects and that the stimulation tolerance of obese subjects appeared to diminish within one EMS session [19]. Similarly, this study observed the physiological responses of the patients in accordance with the electrical impulse intensities. According to our results, the ∆% of body weight in the CON, LIG, MIG, and HIG changed from baseline to −0.74%, −1.34%, −1.28%, and −1.40% at week 4; −0.52%, −1.48%, −1.43%, and −2.04% at week 8; and −0.38%, −2.57%, −5.76%, and −8.88% at week 12, respectively. Similarly, the ∆% of fat mass in the CON, LIG, MIG, and HIG changed from baseline to −1.69%, −5.49%, −6.27%, and −5.95% at week 4; −1.96%, −6.50%, −7.13%, and −6.99% at week 8; and 0.34%, −3.66%, −13.94%, and −28.33% at week 12, respectively. The ∆% of the BMI and percent fat were similar to body weight and fat mass. It can be interpreted that stronger EMS impulse intensities result in decreased body fat.

Unlike the previous studies, Porcari et al. [50] reported that there were no significant changes in the circumferences of the arms or thighs, sum of skinfolds, body weight, percent fat, fat mass, or lean mass between the experimental and control groups after applying EMS. They also reported that the claims relative to the effectiveness of EMS for apparently healthy individuals were not supported by the findings of their studies. These findings may be explained and interpreted as follows. Subsequent stimulation sessions by Porcari et al. [50] were performed three times per week for eight weeks. The areas stimulated during each session were the biceps, triceps, quadriceps, hamstrings, and abdominal muscles. Using such parts of the body for EMS may be problematic, since the sites were no more than a part of the whole body. The second problem was that the electrodes repeatedly detached from the subjects’ skin because of the use of Velcro straps. The third problem was that the number of channels was low, and the fourth problem was that the time off period after EMS was too long. The longer the resting time for muscles after EMS, the longer the time required for the pulse to fall below the threshold value of the muscles to induce muscle contractions again, which may reduce the efficiency of the muscle contractions. Hortobágyi and Maffiuletti [25] suggested that EMS programs that last up to six weeks may induce alterations in the muscle metabolism. Gondin et al. [26,42] and Ruther et al. [27] reported that the techniques of applying EMS for time periods longer than six weeks may cause muscle hypertrophy in the late phases of such programs. Regarding muscle mass, this study found that the ∆% of muscle mass in the CON, LIG, MIG, and HIG changed from baseline to −0.73%, 0.34%, 0.14%, and 0.20% at week 4; −0.67%, 0.73%, 0.61%, and 1.69% at week 8; and −2.21%, 1.38%, 5.31%, and 7.64% at week 12, respectively. In other words, muscle mass showed greater gains as the impulse intensity became higher. A substantial amount of research has also pointed to the positive effects of EMS on the body composition when performed for a period of over 12 weeks [20]. 

This study measured abdominal CT images four times from week zero to week 12 to examine the extent to which WB-EMS affects the abdominal circumference. According to the results, the ∆% of the ASF in the CON changed from baseline to −0.43% at week 4, −1.91% at week 8, and −1.86% at week 12, whereas those of LIG, MIG, and HIG changed from baseline to 0.56%, −0.46%, and −0.44% at week 4; −3.69%, −2.76%, and −2.22% at week 8; and −0.47%, −16.40%, and −25.91% at week 12, respectively. Meanwhile, this study found that the ∆% of the ATF in the LIG, MIG, and HIG-performed isometric exercises combined with WB-EMS changed from baseline to −2.68%, −1.10%, and −1.95% at week 4; −6.44%, −3.65%, and −9.74% at week 8; and −5.21%, −14.99%, and −27.44% at week 12 compared with the ∆% of ATF, which changed from baseline to 1.23% at week 4, −1.76% at week 8, and −0.86% at week 12. These results were similar to those reported by several previous studies, and it was found that the thickness of the abdominal subcutaneous fat can be reliably reduced when wearing WB-EMS and performing isometric exercises. Banerjee et al. [51] confirmed that EMS can be used on sedentary adults to improve physical fitness and may provide a viable alternative to more conventional forms of exercise in this population, as our results and previous studies also suggest.

Psychologically, it is not easy to tolerate exercise with high levels of electrical stimulation. That is, among the three types of WB-EMS intensities applied in this study, we investigated what kind of EMS impulse intensity has the greatest effect on the subjects’ body image and satisfaction. In addition, we looked into whether changes in body image and satisfaction can lead to changes in self-esteem. Originally, body image or body image flexibility was known to be associated with psychological flexibility regarding body image [52]. The BI-AAQ has also been used to measure and evaluate eating disorders, poor psychological health [53,54], distressing thoughts and feelings associated with binge eating [55], anorexia [56], and bulimia [57]. However, this study investigated the psychological changes by using body image questionnaires such as BI-AAQ to measure the body image flexibility after the application of WB-EMS. Body image flexibility is defined as the capacity to experience the ongoing perceptions, sensations, feelings, thoughts, and beliefs associated with one’s body fully and intentionally while pursuing chosen values [30]. Along with the BI-AAQ, this study used the BSQ and SES for observing psychological changes from pretests to posttests. Prior to investigating the effects of WB-EMS on the above variables, Marsh et al. [58] reported that the highest correlations existed between the body and physical appearance factors, with the three correlations relating competence to strength, body, and physical activity. These results indicated that body attractiveness is due to both body traits and physical appearances.

This study assigned the same isometric exercise to young male adults but provided low-, medium-, and high-intensity EMS impulses. We measured the effects of the three types of EMS impulses on the feelings of the participants, as well as the intensities that were most helpful in improving their body image. In addition, this study also used the BSQ to measure psychological satisfaction regarding body image after performing isometric exercises combined with WB-EMS. The results of this study revealed that the ∆% of the BI-AAQ in the CON changed from the baseline to 0.75% at week 4, −0.63% at week 8, and 0.24% at week 12. The ∆% of the BI-AAQ in the LIG changed from the baseline to −4.76% at week 4, 1.11% at week 8, and 2.14% at week 12. The ∆% of the BI-AAQ in the MIG and HIG changed from the baseline to −4.22% and −2.65% at week 4, −5.18% and −7.78% at week 8, and −11.50% and −24.12% at week 12, respectively. Lower BI-AAQ scores indicate higher levels of body image flexibility. In other words, it can be interpreted that the stronger the EMS impulse is, the higher the person’s body image satisfaction. The BSQ showed similar results: −1.45% in the CON, −7.10% in the LIG, −22.95% in the MIG, and −25.09% in the HIG at week 12. Although the ∆% of self-esteem in the CON and LIG changed from the baseline to −3.89% and −4.04% at week 4, −11.62% and −5.41% at week 8, and −13.17% and −7.55% at week 12, those of self-esteem in the MIG and HIG changed from the baseline to 2.25% and 13.25% at week 4, 13.02% and 15.20% at week 8, and 18.81% and 29.68% at week 12. In other words, it can be said that a higher EMS intensity impulse leads to greater physical development and exercise intensity, which can be mentally challenging.

According to some researchers, increased skeletal muscles, improved muscle strength without lifting weights, and even preserving muscle mass can result from the use of EMS [59,60,61]. This combination of both EMS and exercise training can cause additional tension, thus creating more effective results. Gerovasili et al. [60] reported that the electrically induced contractions must be in the range of 60–80% of the maximal voluntary contraction. Based on scientific evidence, this study found that the psychological scores were steadily increased each week, showing that only high-intensity electrical stimulation can improve body image, satisfaction, and self-esteem in healthy men. These findings were supported by the results of studies by Ahmad and Hasbullah [59], Gerovasili et al. [60], and Iwasaki et al. [61]. In addition, this study observed that the BSQ of the HIG showed the lowest score, indicating a very positive result and that the scores for health and physical activity in the HIG were higher than those of the other three groups after 12 weeks. Self-esteem in the HIG also showed a higher tendency compared with those of the other three groups from week zero to week 12.

It can be hypothesized that repeated exposure to WB-EMS training may result in increased physical fitness and muscle function, reduced body fat mass, and improved psychological health. Ahmad and Hasbullah [59] reported that EMS training was able to improve the male body composition. Many studies, including the results from this study, showed that EMS training combined with isometric exercise can decrease fat mass and percent body fat. These results suggest that improved body composition can also increase self-esteem through greater satisfaction with body image and body shape. Similarly, Harvey et al. [62] suggested that the physiological benefits gained from functional electrical stimulation training led to significant psychological benefits as well. Anderson et al. [63] reported that 37 sedentary healthy women participated in baseline testing on a range of anthropometric measures, body composition, and self-perception measures. Subsequently, participants were randomly assigned to one of three groups: walking group, walking + EMS group, or the control group. When comparing the results with the control group after eight weeks, both walking groups had a significant reduction in the number of anthropometric measures and improvements in the self-perception measures. The improvements of both the anthropometric measures and self-perceptions were greater for the walking + EMS group, which indicates that changes in self-perception may be affected by physiological changes.

Compared to the no stimulation EMS control group, all three of the EMS groups exhibited improved tendencies in self-esteem and significant improvements in body image and body shape. This effect was particularly apparent in the mid- and/or high-impulse EMS groups. These results are similar to previous research studies that suggested that exercise enhances self-perception [64,65,66,67,68,69] and is contrary to other studies that have found that exercise does not improve self-perception [70]. Ultimately, this study suggests that a WB-EMS suit equipped with an electrical muscle stimulation device can reduce fat and increase muscle mass, which, in turn, improves the psychological factors. However, this effect only appeared in the EMS group in which a high-impulse-intensity was applied.

## 5. Conclusions

This study confirmed that WB-EMS using high electrical impulse intensity can improve the body composition and psychological factors in healthy male adults. However, our study had some limitations. First, the evaluator was not blinded to the group allocation. Second, the participants consisted entirely of young men with somewhat smaller sizes. Considering these limitations, further studies that investigate the effectiveness of exercising with WB-EMS devices on a greater number of participants with diverse demographic backgrounds are encouraged.

## Figures and Tables

**Figure 1 medicina-57-00191-f001:**
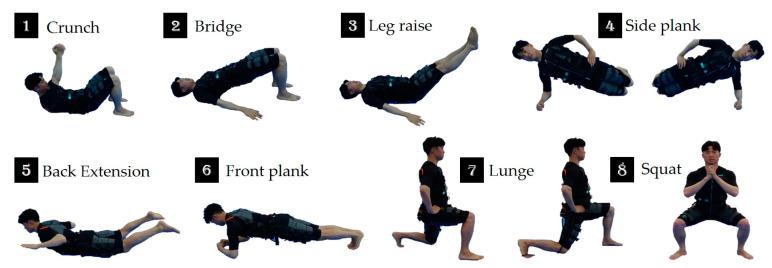
Whole-body electromyostimulation (WB-EMS) exercise program.

**Table 1 medicina-57-00191-t001:** Physical characteristics of the participants.

Items	Groups				Kruskal-Wallis	
	CON (*n* = 13)	LIG (*n* = 13)	MIG (*n* = 14)	HIG (*n* = 14)	*X^2^*	*p*
Age (y)	22.69 ± 1.97	24.31 ± 1.60	23.79 ± 2.08	24.14 ± 1.70	5.026	0.170
Height (cm)	174.62 ± 3.12	175.54 ± 2.63	175.21 ± 2.83	176.71 ± 3.81	2.020	0.568
Weight (kg)	77.15 ± 6.11	77.12 ± 8.26	77.45 ± 7.63	78.49 ± 7.53	0.160	0.984
Muscle mass (kg)	34.40 ± 3.61	35.55 ± 4.14	35.79 ± 4.14	35.59 ± 4.42	1.171	0.760
Fat mass (kg)	14.77 ± 3.48	14.76 ± 3.72	15.14 ± 3.87	15.46 ± 3.52	0.415	0.937
BMI (kg/m^2^)	25.32 ± 2.04	25.05 ± 2.78	25.27 ± 2.75	25.11 ± 2.14	0.213	0.975
Percent fat (%)	21.52 ± 2.85	22.04 ± 2.15	21.99 ± 2.66	21.57 ± 2.81	0.534	0.911

All values are expressed as mean ± standard deviation. BMI, body mass index. CON, control group, LIG, low-impulse-intensity group, MIG, mid-impulse-intensity group, and HIG, high impulse-intensity group.

**Table 2 medicina-57-00191-t002:** Definition and degree of the category scores by the international physical activity questionnaire (IPAQ).

Category		Criteria
#	Activity Degree	
1	Low	Any one of the following 2 criteria - No activity is reported OR - Some activity is reported but not enough to meet Categories 2 or 3.
2	Moderate	Either of the following 3 criteria - 3 or more days of vigorous activity of at least 20 min per day OR - 5 or more days of moderate-intensity activity and/or walking of at least 30 min per day OR - 5 or more days of any combination of walking or moderate or vigorous intensity.
3	High	Any one of the following 2 criteria - Vigorous activity on at least 3 days and accumulating at least 1500 MET/min per week OR - 7 or more days of any combination of walking or moderate- or vigorous-intensity activities, accumulating at least 3000 MET/min per week.

Equations for calculating physical activity degree as follows. Walking MET/min/week = 3.3 × min of activity/day × days per week. Moderate-intensity physical activity MET/min/week = 4.0 × min of activity/day × days per week. Vigorous-intensity physical activity MET/min/week = 8.0 × min of activity/day × days per week. Total MET/min/week = Walking MET/min/week + Moderate-intensity physical activity MET/min/week + Vigorous-intensity physical activity MET/min/week. MET: metabolic equivalent.

**Table 3 medicina-57-00191-t003:** Internal consistencies of the questionnaires.

Times	BI-AAQ		BSQ		SES	
	Items	α	Items	α	Items	α
Week 0	12	0.869	8	0.795	10	0.905
Week 4	12	0.912	8	0.85	10	0.881
Week 8	12	0.937	8	0.874	10	0.818
Week 12	12	0.949	8	0.843	10	0.837

BI-AAQ, Body Image Acceptance and Action Questionnaire, BSQ, Body Shape Questionnaire, and SES, self-esteem.

**Table 4 medicina-57-00191-t004:** Differences of the controlled variables among the four groups.

		Groups				ANOVA (*p*)		
Items	Week	CON	LIG	MIG	HIG	G	T	G*T
Calorie intake (kcal)	0	1566.69 ± 235.08	1576.23 ± 536.80	1599.14 ± 309.87	1588.14 ± 306.0	0.735	0.747	0.716
	4	1616.08 ± 253.59	1459.15 ± 425.58	1664.43 ± 337.68	1630.14 ± 423.88			
	8	1529.23 ± 358.38	1698.77 ± 459.15	1709.71 ± 356.51	1654.71 ± 421.38			
	12	1723.69 ± 417.02	1420.38 ± 390.18	1598.21 ± 402.91	1592.79 ± 247.38			
Calorie output (kcal)	0	376.78 ± 276.88	358.75 ± 195.46	350.16 ± 186.81	369.58 ± 139.28	0.973	0.850	0.991
	4	382.54 ± 192.16	369.40 ± 194.04	345.10 ± 130.88	373.23 ± 192.73			
	8	352.80 ± 231.33	380.33 ± 199.63	363.16 ± 223.14	367.93 ± 250.28			
	12	383.06 ± 314.35	382.19 ± 209.52	360.55 ± 111.42	364.52 ± 83.83			
CK (IU/L)	0	250.69 ± 141.20	266.92 ± 148.20	259.36 ± 73.56	253.57 ± 79.45	0.965	0.215	0.917
	4	261.23 ± 89.64	260.38 ± 60.25	243.86 ± 90.01	258.93 ± 101.45			
	8	269.31 ± 170.18	269.54 ± 86.57	256.93 ± 114.18	279.00 ± 94.91			
	12	265.62 ± 83.93	271.23 ± 89.25	285.36 ± 122.57	292.21 ± 69.64			

All values are expressed as the mean ± standard deviation. CON, control group; LIG, low-impulse-intensity group; MIG, mid-impulse-intensity group; HIG, high-impulse-intensity group; PAC, physical activity category which was scored by 1 (low), 2 (moderate), and 3 (high) activity levels; and CK, creatine kinase. *p*-value was analyzed using the repeated measures ANOVA test. G: group; T: time; G*T: group by time; IU: International unit.

**Table 5 medicina-57-00191-t005:** Changes and differences in the psychological questionnaire.

	Groups					ANOVA (*p*)		
Items	Week	CON	LIG	MIG	HIG	G	T	G*T
BI-	0	61.54 ± 7.36 ^b^	60.25 ± 4.93 ^a^	60.86 ± 4.74 ^a^	61.19 ± 8.77 ^a^	0.117	0.001	0.001
AAQ	4	62.00 ± 7.57 ^a^	57.38 ± 3.55 ^b^	58.29 ± 4.86 ^b^	59.57 ± 9.04 ^a^			
	8	61.15 ± 6.11 ^b^	60.92 ± 4.80 ^a^	57.71 ± 6.39 ^c^	56.43 ± 10.26 ^b^			
	12	61.69 ± 6.38 ^ab^	61.54 ± 7.78 ^a^	53.86 ± 8.69 ^d^	46.43 ± 13.66 ^c^			
BSQ	0	26.23 ± 3.09 ^a^	25.92 ± 3.38 ^a^	26.14 ± 5.20 ^a^	24.79 ± 4.26 ^a^	0.105	0.001	0.001
	4	25.92 ± 3.43 ^a^	25.38 ± 3.04 ^a^	25.36 ± 5.36 ^a^	24.21 ± 3.93 ^a^			
	8	25.85 ± 2.73 ^a^	24.85 ± 3.48 ^b^	23.79 ± 5.01 ^b^	22.21 ± 4.87 ^b^			
	12	25.85 ± 4.86 ^a^	24.08 ± 4.65 ^b^	20.14 ± 5.63 ^c^	18.57 ± 3.61 ^c^			
SEQ	0	29.85 ± 9.37 ^a^	28.46 ± 6.37 ^a^	28.50 ± 9.65 ^b^	29.14 ± 7.95 ^c^	0.081	0.079	0.001
	4	28.69 ± 10.65 ^a^	27.31 ± 5.50 ^a^	29.14 ± 7.68 ^b^	33.00 ± 8.08 ^b^			
	8	26.38 ± 8.14 ^b^	26.92 ± 5.16 ^ab^	32.21 ± 9.50 ^a^	33.57 ± 10.31 ^b^			
	12	25.92 ± 9.73 ^c^	26.31 ± 6.77 ^b^	33.86 ± 7.90 ^a^	37.79 ± 8.04 ^a^			

All values are expressed as mean ± standard deviation. BI-AAQ, Body Image-Acceptance and Action Questionnaire, BSQ, Body Shape Questionnaire, SEQ, self-esteem questionnaire, CON, control group, LIG, low-impulse-intensity group, MIG, mid-impulse-intensity group, and HIG, high-impulse-intensity group. ^a,b,c,d^ Bonferroni post hoc symbols. G: group; T: time; G*T: group by time.

**Table 6 medicina-57-00191-t006:** Changes and differences in the body composition.

	Groups					ANOVA (*p*)		
Items	Week	CON	LIG	MIG	HIG	G	T	G*T
Weight	0	77.15 ± 6.11	77.12 ± 8.26	77.45 ± 7.63	78.49 ± 7.53	0.982	0.001	0.001
(kg)	4	76.58 ± 6.17	76.09 ± 7.66	76.46 ± 7.34	77.39 ± 7.33			
	8	76.75 ± 6.44	75.98 ± 7.73	76.34 ± 7.15	76.89 ± 7.31			
	12	76.86 ± 6.26 ^a^	75.14 ± 6.61 ^a^	72.99 ± 4.85 ^b^	71.52 ± 4.68 ^b^			
Muscle mass	0	34.40 ± 3.61	35.55 ± 4.14	35.79 ± 4.14	35.59 ± 4.42	0.387	0.001	0.002
(kg)	4	34.15 ± 3.92	35.67 ± 4.38	35.84 ± 4.49	35.66 ± 4.66			
	8	34.17 ± 3.73	35.81 ± 4.24	36.01 ± 4.22	36.19 ± 4.38			
	12	33.64 ± 3.92 ^c^	36.04 ± 3.78 ^ab^	37.69 ± 3.42 ^a^	38.31 ± 3.07 ^a^			
Fat mass	0	14.77 ± 3.48	14.76 ± 3.72	15.14 ± 3.87	15.46 ± 3.52	0.946	0.001	0.001
(kg)	4	14.52 ± 3.52	13.95 ± 3.80	14.19 ± 3.96	14.54 ± 3.59			
	8	14.48 ± 3.08	13.80 ± 3.81	14.06 ± 3.63	14.38 ± 3.77			
	12	14.82 ± 3.32 ^a^	14.22 ± 4.09 ^a^	13.03 ± 3.43 ^ab^	11.08 ± 1.80 ^c^			
BMI	0	25.32 ± 2.04	25.05 ± 2.78	25.27 ± 2.75	25.11 ± 2.14	0.801	0.001	0.001
(kg/m^2^)	4	25.15 ± 2.22	24.72 ± 2.68	24.94 ± 2.69	24.78 ± 2.11			
	8	25.19 ± 2.16	24.70 ± 2.69	24.92 ± 2.62	24.61 ± 2.13			
	12	25.23 ± 2.28 ^a^	24.38 ± 2.28 ^a^	23.81 ± 1.98 ^ab^	22.89 ± 1.20 ^c^			
Percent fat	0	21.52 ± 2.85	22.04 ± 2.15	21.99 ± 2.66	21.57 ± 2.81	0.416	0.033	0.024
(%)	4	19.72 ± 2.66	20.08 ± 2.19	19.42 ± 3.86	19.90 ± 3.35			
	8	19.30 ± 2.63	19.02 ± 2.78	18.81 ± 3.66	18.79 ± 3.40			
	12	20.32 ± 1.94 ^a^	19.79 ± 1.27 ^ab^	16.81 ± 2.74 ^c^	15.76 ± 3.09 ^d^			

All values are expressed as the mean ± standard deviation. BMI, body mass index, CON, control group, LIG, low-impulse-intensity group, MIG, mid-impulse-intensity group, and HIG, high-impulse-intensity group. ^a,b,c,d^ Bonferroni post hoc symbols. G: group; T: time; G*T: group by time.

**Table 7 medicina-57-00191-t007:** Changes and differences in abdominal fat.

	Groups					ANOVA (*p*)		
Items	Week	CON	LIG	MIG	HIG	G	T	G*T
Abdominal	0	131.26 ± 84.77	129.55 ± 83.93	118.50 ± 87.18	110.08 ± 76.55	0.681	0.001	0.197
visceral fat	4	133.63 ± 94.24	120.86 ± 94.66	116.20 ± 96.56	105.61 ± 86.90			
(cm^2^)	8	127.82 ± 78.88	116.80 ± 78.35	112.78 ± 81.26	87.83 ± 46.28			
	12	128.40 ± 74.43	115.18 ± 76.99	102.93 ± 62.30	77.51 ± 32.73			
Abdominal	0	151.37 ± 49.83	160.88 ± 50.55	155.18 ± 58.83	153.44 ± 48.88	0.823	0.001	0.001
subcutaneous	4	150.72 ± 49.44	161.78 ± 50.16	154.47 ± 56.53	152.76 ± 48.44			
fat (cm^2^)	8	148.48 ± 46.70	154.94 ± 45.47	150.89 ± 55.02	150.03 ± 48.48			
	12	148.56 ± 45.84 ^ab^	160.13 ± 48.98 ^a^	129.73 ± 38.94 ^c^	113.69 ± 25.13 ^d^			
Abdominal	0	279.37 ± 77.28	290.43 ± 77.50	273.68 ± 92.09	263.52 ± 85.92	0.402	0.001	0.002
total fat	4	282.82 ± 83.10	282.65 ± 80.32	270.66 ± 95.45	258.37 ± 92.23			
(cm^2^)	8	274.44 ± 66.11	271.73 ± 60.05	263.68 ± 82.64	237.86 ± 61.51			
	12	276.96 ± 63.08 ^a^	275.31 ± 66.52 ^a^	232.66 ± 67.29 ^ab^	191.20 ± 34.66 ^c^			

All values are expressed as the mean ± standard deviation. CON, control group, LIG, low-impulse-intensity group, MIG, mid-impulse-intensity group, and HIG, high-impulse-intensity group. ^a,b,c,d^ Bonferroni post hoc symbols. G: group; T: time; G*T: group by time.

## Data Availability

The data presented in this study are available on request from the corresponding author. The data are not publicly available due to ethical restrictions.

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
