# Peer review of "Higher Impulse Electromyostimulation Contributes to Psychological Satisfaction and Physical Development in Healthy Men"

_medicina, 2021, doi:10.3390/medicina57030191_

Round 1

Reviewer 1 Report

In this study, Kim et al. showed the positive effects of whole body-electromyostimulation (WB-EMS) on body composition and psychological responses in healthy male adults. Using the EB-EMS suit and isometric exercise for 12 weeks, they demonstrated that electromyostimulation improve body image and self-esteem, increases muscle mass, and reduce body fat and abdominal fat by pulse intensity- dependent. Based on these findings, authors suggested that high electrical muscle stimulation (WB-EMS suit) training combined with isometric exercise lead to positive effects on body composition and in result, improving psychological responses. Overall, the manuscript is well organized. Methods and study design are clear, conclusions are supported by the results. I have minor points:

Point 1: page 7 – methods – Is there a difference of isometic exercise intensity between wearing a WB-EMS suit device and not wearing it?

Point 2: line 332 – Results – Week in Table 5. Please replace “2” with “12”.

Author Response

Answers to 1st reviewer’s comments

Thank you for your kind advice and comments for publication in Medicina. We revised our manuscript as per your comments. We represented the specific modifications in response to the comments by blue-letters in our manuscript. We sincerely appreciate your comments because your comments make our manuscript better. Details of responses about reviewer’s comments are as follows:

# General Comments or Suggestions

In this study, Kim et al. showed the positive effects of whole body-electromyostimulation (WB-EMS) on body composition and psychological responses in healthy male adults. Using the EB-EMS suit and isometric exercise for 12 weeks, they demonstrated that electromyostimulation improve body image and self-esteem, increases muscle mass, and reduce body fat and abdominal fat by pulse intensity- dependent. Based on these findings, authors suggested that high electrical muscle stimulation (WB-EMS suit) training combined with isometric exercise lead to positive effects on body composition and in result, improving psychological responses. Overall, the manuscript is well organized. Methods and study design are clear, conclusions are supported by the results. I have minor points:

#1. Comments or Suggestions

Point 1: page 7 – methods – Is there a difference of isometic exercise intensity between wearing a WB-EMS suit device and not wearing it?

#1. Response: First of all, thank you for your question. The isometric exercise intensity was the same for all groups. Furthermore, all groups (including the control group) were provided the same EMS-suit, but the control group was not given electrical stimulation. Only LIG, MIG, and HIG were provided different degrees of impulse-intensity. Such contents are listed from Lines 134 to 140, and it may be helpful to refer to the contents below.

“The LIG, MIG, and HIG underwent 20-minute WB-EMS sessions combined with iso-metric exercises in accordance with their intensities of electrical stimulation 3 times a week for 12 weeks. In other words, they received one of three types of electrical stimuli at different intensities according to their maximum tolerance (1MT). The CON also wore the WB-EMS suit as much as the other groups, but they did not receive any electrical stimuli while performing isometric exercises.”

#2. Comments or Suggestions

Point 2: line 332 – Results – Week in Table 5. Please replace “2” with “12”.

#2. Response: Thank you for pointing that out. As you pointed out, “2” which was incorrectly indicated in Table 5 has been modified to “12”. The revised table is as follows.

Table 5. Changes and differences in psychological questionnaire

Groups

ANOVA (p)

Items

Week

CON

LIG

MIG

HIG

G

T

G*T

BI-AAQ

0

61.54 ± 7.36 b

60.25 ± 4.93 a

60.86 ± 4.74 a

61.19 ± 8.77 a

0.117

0.001

0.001

4

62.00 ± 7.57 a

57.38 ± 3.55 b

58.29 ± 4.86 b

59.57 ± 9.04 a

8

61.15 ± 6.11 b

60.92 ± 4.80 a

57.71 ± 6.39 c

56.43 ± 10.26 b

12

61.69 ± 6.38 ab

61.54 ± 7.78 a

53.86 ± 8.69 d

46.43 ± 13.66 c

BSQ

0

26.23 ± 3.09 a

25.92 ± 3.38 a

26.14 ± 5.20 a

24.79 ± 4.26 a

0.105

0.001

0.001

4

25.92 ± 3.43 a

25.38 ± 3.04 a

25.36 ± 5.36 a

24.21 ± 3.93 a

8

25.85 ± 2.73 a

24.85 ± 3.48 b

23.79 ± 5.01 b

22.21 ± 4.87 b

12

25.85 ± 4.86 a

24.08 ± 4.65 b

20.14 ± 5.63 c

18.57 ± 3.61c

SEQ

0

29.85 ± 9.37 a

28.46 ± 6.37 a

28.50 ± 9.65 b

29.14 ± 7.95 c

0.081

0.079

0.001

4

28.69 ± 10.65 a

27.31 ± 5.50 a

29.14 ± 7.68 b

33.00 ± 8.08 b

8

26.38 ± 8.14 b

26.92 ± 5.16 ab

32.21 ± 9.50 a

33.57 ± 10.31 b

12

25.92 ± 9.73 c

26.31 ± 6.77 b

33.86 ± 7.90 a

37.79 ± 8.04 a

All values are expressed as mean ± standard deviation. BI-AAQ, Body-Image-Acceptance and Action Questionnaire; BSQ, Body Shape Questionnaire; SES, self-esteem; CON, control group; LIG, low impulse-intensity group; MIG, mid impulse-intensity group; HIG, high impulse-intensity group. a, b, c and d mean Bonferroni post hoc symbols.

Lastly, there were some typos and minor rewording that was done in the attached manuscript.

Thank you!

Submission Date

February 18, 2021

Reviewer 2 Report

The manuscript is very interesting because besides known training effects of wb-ems training it deals with a very important aspect - psychological effect. The physiological results are not so interesting. Here would recommend to shorten significantly in all sections. Instead, the psychological effects should be more clearly in focus. I find the results section extremely difficult to read, but I know about my own linguistic weakness. It is confusing, and the numerous acronyms make it even more difficult. However, the tables help to understand the data very well. Therefore, I recommend to process only the key message in the text. Even though the sample of subjects is very small and not very representative for medical use. This manuscript allows other research groups to base a study design (case number estimation, etc.). Therefore, I suggest to significant shortening of the text.

Author Response

Answers to 2nd reviewer’s comments

Thank you for your kind advice and comments for publication in Medicina. We revised our manuscript as per your comments. We represented the specific modifications in response to the comments by blue-letters in our manuscript. We sincerely appreciate your comments because your comments make our manuscript better. Details of responses about reviewer’s comments are as follows:

# General Comments or Suggestions

The manuscript is very interesting because besides known training effects of wb-ems training it deals with a very important aspect - psychological effect. The physiological results are not so interesting. Here would recommend to shorten significantly in all sections. Instead, the psychological effects should be more clearly in focus. I find the results section extremely difficult to read, but I know about my own linguistic weakness. It is confusing, and the numerous acronyms make it even more difficult. However, the tables help to understand the data very well.

Therefore, I recommend to process only the key message in the text. Even though the sample of subjects is very small and not very representative for medical use. This manuscript allows other research groups to base a study design (case number estimation, etc.). Therefore, I suggest to significant shortening of the text.

#1. Comments or Suggestions

Here would recommend to shorten significantly in all sections. Instead, the psychological effects should be more clearly in focus. I find the results section extremely difficult to read, but I know about my own linguistic weakness. It is confusing, and the numerous acronyms make it even more difficult. However, the tables help to understand the data very well.

#1. Response: Thank you for what you pointed out. According to your suggestion, the overall contents of our manuscript have been organized as follows to make it easier to understand. Moreover, we tried to shorten our text as follows.

On Lines 310 to 315:

As shown in Table 1, there were no significant differences in age (p = 0.170), height (p = 0.568), and weight (p = 0.984) among the four groups. There were also no significant differences in muscle mass (p = 0.760), fat mass (p = 0.937), BMI (p = 0.975), and percent fat (p = 0.911) among the four groups. The demographic variables of this study indicated homogeneity of subjects. Table 4 describes the calorie intake, calorie output, and CK among the four groups during the experimental period. There were no significant differences in the controlled variables of this study.

On changed Lines 310 to 313:

“As shown in Table 1, there were no significant differences among the four groups. The demographic variables of this study indicated homogeneity of subjects. There were also no significant differences in the controlled variables, as shown in Table 4.”

On Lines 319 to 326:

As shown in Table 5, there were significant interaction effects for all psychological questions. Three psychological scales in CON showed negative changing tendencies, whereas those in LIG, MIG, and HIG showed positive changing tendencies compared to those of CON. The ANCOVA revealed that BI-AAQ (F = 5.017, p = 0.004), BSQ (F = 4.680, p = 0.007), and SEQ (F = 8.468, p = 0.001) were significantly different among the four groups. In particular, the higher the impulse intensity, the greater the positive change. In other words, HIG, which received the highest impulse intensity showed the most improved value in Week 12, which was confirmed by the Bonferroni post hoc test.

On changed Lines 319 to 325:

“There were significant interaction effects for all psychological questions (Table 5). Three psychological scales in CON showed negative changing tendencies, whereas those in the other groups showed positive changing tendencies. The ANCOVA revealed that BI-AAQ (F = 5.017, p = 0.004), BSQ (F = 4.680, p = 0.007), and SEQ (F = 8.468, p = 0.001) were significantly different among the four groups (not shown in Table 5). In particular, HIG showed the most improved value in Week 12, which was confirmed by the Bonferroni post hoc test.”

On Lines 332 to 346:

As shown in Table 6, baseline variables in the subjects of the four groups showed homogeneity. Body weight, fat mass, BMI, and percent fat in CON showed decreasing tendencies, whereas those in LIG, MIG, and HIG showed a noticeable decrease compared to those of CON. There were significant interactions in all variables in the repeated ANOVA test. However, the ANCOVA test revealed that although body weight (F = 6.354, p = 0.001), fat mass (F = 7.368, p = 0.001), and BMI (F = 6.427, p = 0.001) in three groups that received electrical stimulation were significantly lower than those in CON, the percent fat (F = 2.268, p = 0.092) did not show a significant difference among groups. For body weight, fat mass, and BMI, higher the impulse intensities led to greater decreases. In other words, HIG, which received the highest impulse intensity, showed the lowest value in Week 12, which was confirmed by the Bonferroni post hoc test. However, muscle mass showed a different tendency compared with the above variables. The ANCOVA revealed that muscle mass (F = 5.758, p = 0.002) in the three groups that received electrical stimulation were significantly higher than those in CON. In particular, the higher the impulse intensity was applied, the greater the increase resulted.

On changed Lines 319 to 340:

“As shown in Table 6, weight, fat mass, BMI, and percent fat in CON showed decreasing tendencies, whereas those in LIG, MIG, and HIG showed a noticeable decrease. There were significant interactions in all variables in the repeated ANOVA test. The ANCOVA test revealed that although weight (F = 6.354, p = 0.001), fat mass (F = 7.368, p = 0.001), and BMI (F = 6.427, p = 0.001) in the three experimental groups were significantly lower than those in CON, the percent fat (F = 2.268, p = 0.092) did not show a significant difference. Muscle mass showed a different tendency and the ANCOVA revealed that muscle mass (F = 5.758, p = 0.002) in the experimental three groups was significantly higher than those in CON. In particular, the higher the impulse intensity was applied, the greater the increase of muscle mass resulted.

On Lines 347 to 355:

As shown in Table 7, although there was no interaction effect in the abdominal visceral fat area, there were significant interactions in the abdominal subcutaneous fat and total fat areas. Both abdominal subcutaneous fat and total fat in CON showed decreasing tendencies, whereas those in LIG, MIG, and HIG showed a noticeable decrease compared to those of CON. The ANCOVA revealed that abdominal subcutaneous fat (F = 5.517, p = 0.002) and abdominal total fat (F = 10.933, p = 0.001) were significantly different among the four groups. In particular, the higher the impulse intensity was applied, the greater the decrease was. In other words, HIG, which received the highest impulse intensity showed the lowest value in Week 12, which was confirmed by the Bonferroni post hoc test.

On changed Lines 319 to 353:

“Although there was no interaction effect in the abdominal visceral fat area, there were significant interactions in the abdominal subcutaneous fat and total fat areas (Table 7). Both subcutaneous fat and total fat in CON showed decreasing tendencies, whereas those in the experimental groups showed a noticeable decrease. The ANCOVA revealed that subcutaneous fat (F = 5.517, p = 0.002) and total fat (F = 10.933, p = 0.001) were significantly different among groups. In particular, HIG showed the lowest value in Week 12, which was confirmed by the Bonferroni post hoc test.”

#2. Comments or Suggestions

…Even though the sample of subjects is very small and not very representative for medical use. This manuscript allows other research groups to base a study design (case number estimation, etc.).

#2. Response: Thank you for what you pointed out. In this study, subjects were assigned after calculating and deriving the entire sample using the G*power program before the start of the study. We would appreciate it if you can refer to the information below.

On Lines 291 to 296:

“The sample size was determined using G*Power v. 3.1.9.7 [47,48], considering a priori effect size of f2 (V) = 0.25 (medium size effect), α error probability = 0.05, power (1-β error probability) = 0.95, number of groups = 4, and number of measurements = 4. There were 13 to 14 subjects that were assigned to each of the 4 groups, with a total of 52 subjects based on the numbers assigned to this program.”

Lastly, there were some typos and minor rewording that was done in the attached manuscript. Thank you!

Submission Date

February 18, 2021
